# Acute Vision Loss as the Initial Manifestation of Granulomatosis with Polyangiitis Involving the Orbital Apex

**DOI:** 10.3390/diagnostics12071540

**Published:** 2022-06-24

**Authors:** Shang-Yen Wu, Ling-Shuo Chang, Yuan-Chieh Lee, Yu-Jen Pan, Yu-Fu Chou, Fang-Ling Chang, Yu-Hsuan Lu, Nancy Chen

**Affiliations:** 1Department of Ophthalmology, Hualien Tzu Chi Hospital, Buddhist Tzu Chi Medical Foundation, Hualien 970, Taiwan; sun90602@hotmail.com (S.-Y.W.); u101022001@gmail.com (L.-S.C.); yuanchieh.lee@gmail.com (Y.-C.L.); fangling0103@gmail.com (F.-L.C.); yuhsuan810@gmail.com (Y.-H.L.); 2Department of Ophthalmology and Visual Science, Tzu Chi University, Hualien 970, Taiwan; 3Institute of Medical Sciences, Tzu Chi University, Hualien 970, Taiwan; 4Division of Allergy, Immunology & Rheumatology, Hualien Tzu Chi Hospital, Buddhist Tzu Chi Medical Foundation, Hualien 970, Taiwan; krannpan@gmail.com; 5School of Medicine, Tzu Chi University, Hualien 970, Taiwan; 6Department of Otolaryngology, Hualien Tzu Chi Hospital, Buddhist Tzu Chi Medical Foundation, Hualien 970, Taiwan; yufuchou@yahoo.com.tw

**Keywords:** granulomatosis with polyangiitis, vision loss, paranasal sinusitis, PR3-anti-neutrophil cytoplasmic antibody

## Abstract

Granulomatosis with polyangiitis (GPA) is a systemic autoimmune disease consisting of necrotizing granulomatosis of the respiratory tract, necrotizing vasculitis, and necrotizing glomerulonephritis. It is under the category of ANCA-associated vasculitis, which involves small vessels. The nose, sinus, and ear were the most affected sites besides lung and kidney in localized form. They might precede other disease manifestations before progressing to the systemic form. Our patient presented with an intractable headache, followed by acute vision loss. His symptoms deteriorated regardless of antibiotic treatment for paranasal sinusitis. The sequential CT/MRI images showed the inflammatory raid of the orbital apex and cavernous sinus within days. The sinus biopsy and elevated PR3-anti-neutrophil cytoplasmic antibody led us to the diagnosis of GPA. Fortunately, the patient’s vision improved gradually after steroid and immunosuppressant treatment.

**Figure 1 diagnostics-12-01540-f001:**
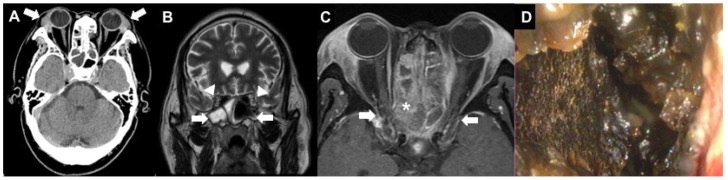
A 63-year-old male farmer presented to the emergency department with severe headache, nasal obstruction, and blurred vision for one month. Ocular examination showed bilateral periorbital edema and swelling of the lacrimal gland. The corrected visual acuity (CVA) was 20/40 in the right eye and 20/200 in the left eye. There was no limitation in all directions of extraocular movements. No relative afferent pupillary defect (RAPD) sign was detected. Non-contrast brain computerized tomography (CT) revealed enlarged lacrimal glands (**A**, arrows) and opacification in paranasal sinuses (**B**, arrows) without optic canal compression (**B**, arrowheads) in the orbital magnetic resonance imaging (MRI). He received systemic intravenous antibiotics for sinusitis. However, three days later, the patient’s vision deteriorated to only light sensation with a positive RAPD sign in the right eye. Orbital MRI with contrast showed polypoid lesions and fluid collection at the sinuses (**C**, asterisk) with inflammation of cavernous sinus and bilateral orbital apexes (**C**, arrows). Emergent functional endoscopic surgery and bilateral lamella papyracea removal to the optic canal were performed. Lots of necrotic tissue (**D**) was found. After the surgery, all the periorbital pain and headache improved remarkably. Later on, laboratory tests revealed an elevation of erythrocyte sedimentation rate and PR3-anti-neutrophil cytoplasmic antibody (c-ANCA). There was no bacterium or fungus infection from the culture reports. A systemic survey revealed that the patient had no active lung lesion, cough, hemoptysis, or shortness of breath. His renal function and urine tests were within normal limits during the follow-up period. However, he had sinusitis with necrotic discharge but did not have deafness or epistaxis. He did not experience skin rash or dermatitis.

**Figure 2 diagnostics-12-01540-f002:**
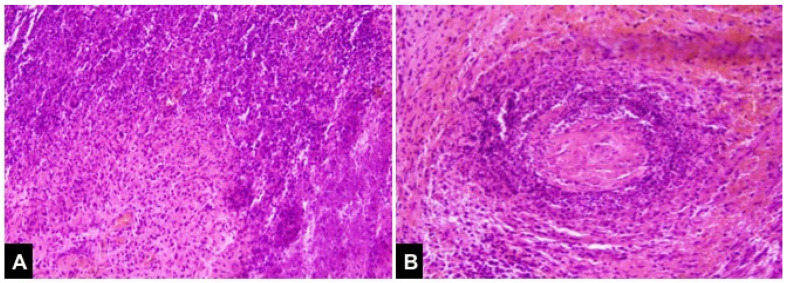
The pathological result demonstrated granulomatous inflammation, geographic necrosis (**A**), and vasculitis (**B**), which were compatible with granulomatosis with polyangiitis. Prednisolone was started as soon as the c-ANCA was found positive at a dosage of 40 mg/day for a month, then tapered gradually with the conjunction of methotrexate use. His vision improved initially but worsened again in the second month despite his cANCA decreasing from 37.2 IU/mL to 4.8 (normal 3.0). Thus, pulse therapy with intravenous methylprednisolone 1000 mg for 3 days and intravenous cyclophosphamide 1000 mg was administrated in the second month since the initial presentation. Another two courses of cyclophosphamide were administrated in the third and fourth months, each with a dosage of 1300 mg. However, his symptom fluctuated as well as his serum ANCA. He thus received rituximab therapy with two doses of 1 gm rituximab, each 14 days apart, followed by maintenance therapy with azathioprine. His cANCA level decreased gradually and turned negative 3 months later after rituximab.

**Figure 3 diagnostics-12-01540-f003:**
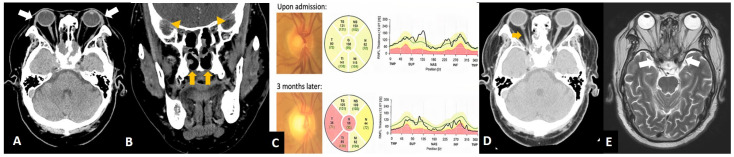
During outpatient clinic follow-up after 3 months, orbital CT showed no dacryoadenitis (**A**, arrows). The sinus was clean (**B**, arrows), and there was no optic nerve compression. (**B**, arrowheads). Nevertheless, fundus photos at post-operative 3 months showed pale disc and retinal nerve fiber layer loss in the right eye (**C**, arrow). Upon the 7-month follow-up, optic atrophy could be noticeable in the CT (**D**, arrow) and the inflammation of the orbital apex and cavernous sinus was quiescent in the MRI image (**E**, arrows). Eventually, his visual acuity was 20/50 in both eyes at one year mark. Granulomatosis with polyangiitis (GPA), known as Wegener’s granulomatosis previously, is an autoimmune vasculitis notably involving the respiratory tract and the kidney. Involvement of the upper respiratory tract accounts for 90% of cases and can cause symptoms of rhinitis, sinusitis, deafness, or epistaxis [1]. Ocular symptoms presented in 30–52% of patients with GPA [1,2] and caused vision loss in 8–37% [1,3]. GPA might cause granulomatous inflammation or vasculitis involving almost any eye structure, resulting in tarsal–conjunctival diseases, keratitis, episcleritis, scleritis, and uveitis [4]. Orbital granuloma accounts for 45% of ocular GPA presentation in the later stage [5]. Our patient did not manifest intra-orbital granulomas, but he suffered from orbital apex complications caused by severe necrotizing sinusitis. The local orbital GPA, if occurred, was usually accompanied by paranasal sinus diseases and bone erosion [6]. Through image studies, the destruction occurred not only at the orbital wall adjacent to the granuloma but also in the nasal septum and cartilage of the nose, suggesting that, besides the direct mass effect, the underlying local inflammation was responsible for the destructive process as well [7]. Optic nerve involvement may manifest as compressive, ischemic, or inflammatory optic neuropathy [7,8,9], frequently resulting in devastating vision loss. The current study assumed that the acute vision loss in the right eye was caused by granuloma compression at the optic canal and local inflammation contiguous with paranasal sinuses. Likewise, lower cranial nerve palsy attributed to sinusitis extension to the jugular foramen was reported in a GPA relapse case [10]. Treatment of GPA is based on severity of the disease. For patients with non-organ- and non-life-threatening diseases, such as sinusitis, initial therapy with glucocorticoids combined with methotrexate is usually adequate. In cases with life-threatening diseases, glucocorticoids extend the life expectancy of GPA from less than 20% to 34% in the 1-year survival rate [4]. Because of poor prognosis, the standard treatment for organ-threatening or life-threatening disease consists of an induction therapy with both cyclophosphamide and glucocorticoid and a maintenance therapy of azathioprine for less toxicity [4,7]. The pathogenesis of antineutrophil cytoplasmic antibody (ANCA)-associated vasculitis diseases implied the significant role of ANCAs, produced by plasma cells, and belong to the lineage of B-lymphocyte. By targeting the B-lymphocyte, cyclophosphamide effectively induces the remission of GPA but with significant morbidity and the worries of infertility. Rituximab, a monoclonal antibody against CD-20 antigen, proved comparable to cyclophosphamide in remission induction [11]. It is also effective to reinduce remission in ANCA-associated vasculitis relapse patients with a better safety profile [12]. Therefore, with corticosteroids, rituximab is established as a practical induction regimen. In the study of Holle et al., 19% of GPA patients with orbital mass became blind even under intense immunosuppression, whereas a longer time to remission was a significant risk factor [7]. In conclusion, our study illustrated that GPA should be kept in the differential diagnosis when encountering patients with drug-resistant sinusitis and vision loss, even without anterior segment symptoms or obvious orbital lesions. Besides, the inflammatory etiology must be considered in a fulminant presentation mimicking infection; the multidisciplinary specialties should all team up to solve the clinical puzzle and treat patients in time.

## Data Availability

Not applicable.

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
