# Peer review of "Acute Vision Loss as the Initial Manifestation of Granulomatosis with Polyangiitis Involving the Orbital Apex"

_diagnostics, 2022, doi:10.3390/diagnostics12071540_

Round 1

Reviewer 1 Report

1-    This manuscript describes  a case  of Wegener granulomatosis occurring in  the orbital cavity as a consequence of sinusytis of the cavernous sinus with impairment of optic nerves. There is a lot body of evidence on multiple cranial nerves palsy due to this granulomatosis with polyangitis. Due to this location , corresponding images with MRI and endoscopic findings give an easy learning of the severity of this disease.

2-    It should be specified the autoimmune nature of this disease and that the current approach is changed due to the anti-CD20 therapies ( Ritiximab)  

3-    It should be emphasized in the discussion that cranial nerves are frequently interested in case  of symptoms like sinusytis .  An interesting case report is reported in the paper : DOI:10.3892/mco.2018.1748 that could be added in the reference

Best regards 

Dr. G Lazzari

Author Response

 Point 1: It should be specified the autoimmune nature of this disease and that the current approach is changed due to the anti-CD20 therapies (Rituximab)  

 Response 1:

The pathogenesis of AAV(antineutrophil cytoplasmic antibody-associated vasculitis) diseases implied the significant role of ANCAs, which were produced by plasma cells, and belong to the lineage of B-lymphocyte. By targeting the B-lymphocyte, cyclophosphamide effectively induces the remission of GPA but with significant morbidity and the worries of infertility. Rituximab, a monoclonal antibody against CD-20 antigen, proved comparable to cyclophosphamide in remission induction[1]. It is also effective to reinduce remission in AAV relapse patients with a safer profile[2]. Therefore, in conjunction with corticosteroids, rituximab is established as an induction strategy.     

Point 2: It should be emphasized in the discussion that cranial nerves are frequently interested in case of symptoms like sinusytis. An interesting case report is reported in the paper: DOI:10.3892/mco.2018.1748 that could be added in the reference.

Response 2: Thanks for the very constructive suggestion. The c-ANCA negative GPA flare-up article described a devastating relapse course after ten years of immunomodulatory treatment. It was noteworthy that the sinusitis extension resulting from GPA could cause low cranial nerve palsies[3]. We would cite the reference in our work.  

  1. Stone, J.H.; Merkel, P.A.; Spiera, R.; Seo, P.; Langford, C.A.; Hoffman, G.S.; Kallenberg, C.G.; St Clair, E.W.; Turkiewicz, A.; Tchao, N.K.; et al. Rituximab versus cyclophosphamide for ANCA-associated vasculitis. N Engl J Med 2010, 363, 221-232, doi:10.1056/NEJMoa0909905.
  2. Smith, R.M.; Jones, R.B.; Specks, U.; Bond, S.; Nodale, M.; Aljayyousi, R.; Andrews, J.; Bruchfeld, A.; Camilleri, B.; Carette, S.; et al. Rituximab as therapy to induce remission after relapse in ANCA-associated vasculitis. Ann Rheum Dis 2020, 79, 1243-1249, doi:10.1136/annrheumdis-2019-216863.
  3. Lazzari, G.; Briatico Vangosa, A.; Assunta De Cillis, M.; Buccoliero, G.; Silvano, G. Lower cranial nerve palsy during radiotherapy for glottic cancer in a patient with Wegener's granulomatosis: An interesting case report. Mol Clin Oncol 2019, 10, 147-152, doi:10.3892/mco.2018.1748.

Reviewer 2 Report

A case description of granulomatosis with polyangiitis. I have no major comments. I propose to add to the description:

-          Occurrence or absence of other (non-ocular) symptoms and signs (pulmonary, renal, cutaneous, systemic, etc.)

-          To specify the doses of cyclophosphamide and glucocorticoids (the term standard dose is sometimes understand in different way)

Author Response

Dear reviewer,

Thanks for the constructive commands and suggestions. I’ve revised the manuscript accordingly; in the following are the related details.

Point 1: Occurrence or absence of other (non-ocular) symptoms and signs (pulmonary, renal, cutaneous, systemic, etc.)

Response 1:
Regarding the occurrence or absence of other (non-ocular) symptoms and signs (pulmonary, renal, cutaneous, systemic, etc.), the patient had no active lung lesion, no cough, hemoptysis, and shortness of breath. His renal function were urine tests were within normal limits during the follow-up period. However, he had sinusitis with necrotic discharge but did not have rhinitis, deafness, or epistaxis. He did not experience skin rash or dermatitis.

Point 2: To specify the doses of cyclophosphamide and glucocorticoids (the term standard dose is sometimes understand in different way)

Response 2:
Regarding the dosage of cyclophosphamide and glucocorticoids, cyclophosphamide was administrated three times intermittently 3-4 weeks apart in the second to the fourth month since the initial presentation, each time with a dosage of 1000 mg; prednisolone was started as soon as the c-ANCA found positive with a dosage of 40 mg/day for a month, then tapered gradually with the adjunction of immunomodulators use.

This manuscript is a resubmission of an earlier submission. The following is a list of the peer review reports and author responses from that submission.